# Why Do Nanoparticles (CNTs) Reduce the Glass Transition Temperature of Nanocomposites?

**Gad Marom** 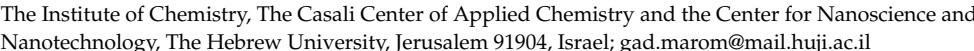

The Institute of Chemistry, The Casali Center of Applied Chemistry and the Center for Nanoscience and Nanotechnology, The Hebrew University, Jerusalem 91904, Israel; gad.marom@mail.huji.ac.il

**Abstract:** This 'opinion' article has been undertaken to provide a plausible answer to the question of why nanocomposites that are reinforced by acicular nanoparticles such as carbon nanotubes (CNTs) do not exhibit the anticipated physical properties—particularly, why the glass transition temperature in some compositions exhibits huge decreases, contrary to expectations. It is claimed that this behavior is typical of fully exfoliated, uniformly dispersed nanocomposites, whose structure is that of molecular composites or solid solutions, and which abide by colligative rules.

**Keywords:** nanocomposites; glass transition temperature; carbon nanotubes; solid solutions; molecular composites

## 1. Introduction

A fundamental characteristic of amorphous and semi-crystalline polymers is to exhibit a second-order transition, which is designated as the 'glass transition', wherein at a specific temperature, the polymer changes from glassy to rubber-like. The glass transition temperature ($T_g$) is the temperature at which sufficient free volume has developed in the polymer to allow for chain segment mobility, which is expressed by turning a hard, glassy polymer to soft and rubbery. Accordingly, the factors that affect chain mobility, e.g., cross-linking, crystallinity, and fillers, affect, in turn, the $T_g$, so that higher levels of cross-linking, crystallinity, or filler loadings result in higher $T_g$ values. Thus, polymeric composite materials in general and nanocomposites specifically are usually expected to generate increases in the glass transition temperature, relative to that of the pristine matrix. This phenomenon is based on the expected effect of the reinforcement and the fillers on the internal structure of the amorphous matrix. This effect works to reduce the free volume and to limit chain segment movement. An additional hindrance to chain movement can result from reinforcement-matrix interfacial interactions, wherein chain segments may be pinned to the surface of the reinforcement (see, e.g., Dibenedetto's classical article [1]). Surprisingly, as discussed below, cases have been reported in the literature where the glass transition temperature of nanocomposites exhibits significant decreases. As we began to explore this anomaly, it occurred to us that it could be related to the more general question of why nanocomposites that are reinforced by acicular nanoparticles such as carbon nanotubes (CNTs) do not exhibit the theoretically predicted extremely high properties. In a recent review article [2], we showed that, although properties such as the modulus of elasticity of CNT-reinforced polymer composites follow rule-of-mixtures straight lines, the modulus values fall significantly short of the predicted ones, which are based on the superb properties of CNTs. This situation is observed even under conditions of highly exfoliated, uniformly dispersed CNTs, whose actual aspect ratio is above the critical value, and which supposedly should exhibit an effective stress transfer mechanism.

In its conclusions, the review article suggested a number of take-home messages. It pointed out that, despite an unprecedented interest in harnessing CNTs for composite materials with ultra-high mechanical properties, the expected extremely high mechanical

properties with even small concentrations, as predicted by conventional micromechanical models, have failed to materialize. Indeed, a number of examples exist wherein the nanoparticles are supposedly well dispersed and not agglomerated, and where, for low nanoparticle contents, an apparent linear behavior of the strength and modulus is observed. However, in most of the examples, the empirical values fall significantly below the predicted ones, which suggests that the stress transfer mechanism in those nanocomposites is insufficiently effective, and that the nanofibers do not realize their full potential contribution. In view of these observations, we then hypothesized that the question of whether conventional micromechanical models apply to nanocomposites relates to the structural differences between composite and nanocomposite materials. Where the first is characterized by a distinct interface between the matrix and the reinforcement, rendering it a heterogeneous material, the latter is a quasi-homogeneous molecular blend, similar to a solid solution.

In this 'opinion' article, we wish employ an inclusive perspective and to show that the same structural differences between nano and micro composites, which affect the mechanical behavior of nanocomposites, account for the behavior of their glass transition temperature, and can explain cases in which nanoparticles generate reductions in the glass transition temperature.

## 2. Discussion: The Glass Transition of Nanocomposites

Generally, the glass transition temperature ($T_g$) of a nanocomposite is expected to be higher than that of the pure matrix material, and this is known as the $T_g$ enhancement. When this occurs, it is ascribed to the presence of nanoparticles, which act as physical cross-links, increasing the rigidity of the matrix. The $T_g$ enhancement effect is more pronounced in cases of smaller nanoparticles and a higher loading. Indeed, it is broadly documented in the literature that the polymers reinforced with either fibers or particles exhibit significant increases in the $T_g$ of the pristine polymer matrix, which is attributed to a decrease in the free volume and to constraints of the chain mobility [1]. This effect is similar to that of crystallinity in a semi-crystalline polymer, where the crystalline regions limit the chain mobility and the $T_g$ goes up with the degree of crystallinity [3]. For CNTs specifically, it has been claimed, for example, that in MWCNT/epoxy composites (where MW stands for multi wall), that increasing concentrations of MWCNTs, as well as functionalizing MWNCTs, leads to an increase in the glass transition temperature, with a higher interfacial interaction between the CNT and the polymer matrix [4].

In some cases, however, in addition to the anomaly in the mechanical performance of nanocomposites, as discussed above, their physical properties and the glass transition temperature do not conform with expected composite behavior. In nanocomposites, significant decreases in the $T_g$ are often recorded, including in cases of uniformly dispersed, exfoliated nanoparticles. For example, the $T_g$ of high-impact polystyrene (HIPS)/TiO$_2$ nanocomposites containing 1–3 wt% TiO$_2$ is slightly lower than that of HIPS. It is claimed that the lowering of the $T_g$ and the occurrence of extra free volume results from the enhanced segmental motion of the HIPS molecules located in the interface region [5]. In another example, a study on the effect of hybrid nanoparticles on the $T_g$ of polymer nanocomposites demonstrates that the experimentally observed glass transition temperature may be either above or below that of the polymer matrix, depending on the size of the nanoparticles [6].

The empirical observations of lowering the glass transition temperature of polymer matrices with nanoparticles have been sustained recently by a study—based on atomistic molecular simulations—on the effect of nanofiller dispersion on the glass transition behavior of cross-linked epoxy-carbon nanotube (CNT) nanocomposites [7]. It was reported that nanocomposites containing dispersed CNTs showed a huge depression in their glass transition temperature, by ∼66 K as compared to the neat cross-linked epoxy. It is also reported that such a large depression was not observed in the nanocomposite containing aggregated CNTs.

In a sequential article published by the same group [8], a suggested explanation for the observations in reference [7] focused on the poor interfacial interactions between the CNTs and the cross-linked epoxy matrix. Under such conditions of poor interfacial interactions, which result in a low particle–matrix bond strength and an ineffective stress transfer mechanism, mechanical properties exhibit low values. Specifically, they showed that, in spite of the presence of stiff CNTs in the nanocomposite, the Young's modulus of the nanocomposite that contained dispersed pristine CNTs did not exhibit any improvement compared to the neat cross-linked epoxy matrix. However, when amido-amine functionalized CNTs were used, the mechanical reinforcement due to the CNTs became more effective, resulting in a ∼50% increase in the Young's modulus compared to the neat cross-linked epoxy—though this was still much below their potential contribution. Concomitantly, a recovery in the $T_g$ was also observed, making it effectively the same as that of the neat polymer. As was claimed, the results demonstrated that the functionalization of the CNTs facilitated the interfacial bond strength and the transfer of mechanical load across the matrix–particle interface.

So why is it then, that fully exfoliated, surface untreated CNTs generate a depression in the $T_g$ that disappears upon treating them to form a strong interfacial nanoparticle–matrix bond? A partial explanation may be found in an article entitled: "How carbon nanotubes affect the cure kinetics and glass transition temperature of their epoxy composites?—A review" [9]. Referring to the $T_g$, the authors present the different effects that unmodified MWCNT and SWCNT (single wall carbon nanotubes) have on the $T_g$ of their epoxy composites. They claim that, where SWCNT may lead to a decrease in the $T_g$ due to its bundling tendency, the results reported for MWCNT suggest an increased or unchanged $T_g$ of the composites. Their given explanation was that the differences correspond to structural ones, which stem—in the case of the epoxy resin—from cure variations inspired, in part, by the CNTs.

We first related to the structural issue of nanocomposites in a recent viewpoint article [10], where we responded to a 1992 *News and Views* article in *Nature* [11], which suggested that nanocomposites, such as carbon nanotube filled polymers, form a new class of fine-scale composites, and then raised the question—in view of the absence of relevant empirical data at that time—of how they would perform mechanically in comparison with classical micrometer-scale composites. The main claim of our viewpoint article—written 25 years thereafter—was that nanocomposites with fully exfoliated, uniformly dispersed nanoparticles are structurally different from microcomposites, and that they form solid solutions that could behave like molecular composites. Their properties are controlled by the chemo-physical interactions on the molecular scale between the nanoparticles and the polymer matrix, and if so, they should also be reflected in the physical properties of the nanocomposites. A consecutive article then presented a conclusive hypothesis that, like (solid) solutions, nanocomposites are expected to exhibit colligative properties, where the effective parameter in determining a physical property, such as the glass transition temperature, is the ratio of the number of solute particles (CNTs) to the number of solvent (polymer) molecules in a (solid) solution. It can then be envisaged—as for lowering the melting point of crystalline materials—that a higher number of nanoparticles will result in a larger decrease in the $T_g$ [12].

In contrast, in cases of strong interfacial bonding, as described above for the amido-amine functionalized CNTs [7], they supposedly react chemically with the matrix. Under such conditions, a new assembly of continuous molecular chain networks is formed, which integrates the CNTs into a structure that is different from that of a solid solution.

Perceptibly, this new structure should not abide by the composites' micromechanics rules that draw on an effectively active stress transfer mechanism; for three-dimensionally unoriented, evenly dispersed CNTs, the mechanical properties would be isotropic and given by a weighted average of the constituents, and the unique mechanical properties of CNTs would not be realized.

In summary, based on the discussion above, and specifically on the simulations of Kahre et al. [7,8], we can refine our earlier definition of nanocomposites as solid solutions, where a clear distinction is now made between two systems, depending on the bonding between the nanoparticles and the matrix. The first, a solid solution, comprising uniformly dispersed nanoparticles that form weak or no interfacial interactions with the polymer matrix. The second, a molecular composite, in which the nanoparticles form strong interfacial bonding with the polymer and integrate into a network of continuous molecular chains.

On a side track to this opinion article, and bearing in mind the low mechanical performance of nanocomposites relative to the expectations, I anticipate that the predominant interest in CNT-reinforced nanocomposites would shift from mechanical properties and engineering applications—which are unachievable—to their unique physical properties due to the CNTs. Lists of potential and actual applications already exist within the literature, which include utilizations of CNTs in energy storage, water filters, thin-film electronics, coatings, actuators, electromagnetic shields, and for drug delivery and biosensing platforms. Such applications, by and large, require the embedding of the CNTs in a polymer matrix to form a nanocomposite. A number of novel examples that shift the focus from mechanical to physical properties already exist in the literature, of which I chose to cite examples from the work of Kim, J.K. [13,14]. Additionally, if the perception of nanocomposites as solid solutions/molecular composites is proven and accepted, a new science of nanocomposites can be anticipated. This new science will relate to the unique physical properties of different nanoparticles (e.g., electrical and thermal conductivities, adsorption capabilities, and chemical activity, etc.), to generate a wealth of material combinations with unique properties, for a wide range of applications.

## 3. Conclusions

The conclusions of this opinion article are based on the realization that different nanoparticles, their level of exfoliation and type of dispersion, and their interaction with the polymer matrix, determine the nanometric/molecular structure of nanocomposites. Specifically, a distinction is made between two structures, namely, a solid solution and a molecular composite, depending on the bonding between the nanoparticles and the matrix. For weak or no interfacial interactions, a solid solution is obtained, whereas for strong interfacial bonding, where a network of continuous molecular chains is formed, the structure is considered to be a molecular composite.

**Funding:** This research received no external funding.

**Data Availability Statement:** The data presented in this study are listed in the References, which are openly available in the web.

**Conflicts of Interest:** The author declares no conflict of interest.

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
