# Peer review of "Why Do Nanoparticles (CNTs) Reduce the Glass Transition Temperature of Nanocomposites?"

_jcs, doi:10.3390/jcs7030114_

Round 1

Reviewer 1 Report

The opinion paper by Marom provides a very wide scope on the Tg change in nanocomposites.

It is asked why a depression in Tg was recorded for well-exfoliated CNT in such composites.

Some of the reported reduced Tg values in the ms were obtained by simulations only (ref 5), as well as the indication of NT-matrix covalent bonding. The text does not mention that this is a simulation result, and it should. There are various papers (e.g., Pötschke) that contradict these observations.

Since the matrix-filler bonding is at the center of this opinion ms (L101-5), a much broader scope on relevant systems should be given (not only the amido-Amine functionalization-epoxy) to show that the concept is general.

Regarding the colligative approach of the mechanical properties in composites mentioned in this ms: it has been shown in so many papers that most properties are almost always way below the predicted by the rule of mixture. I cannot see the relevance of such a statement to this ms.

In summary, the author keeps referring to his/her older papers, and one wonders where is the new opinion here? It is vaguely mentioned only in the last few lines of the ms. What are the other physical properties the author refers to? This is not clear at all.

Author Response

I wish to thank the reviewer for his/her highly useful comments that helped me upgrade the manuscript. Below is my response to his comments point by point:

Some of the reported reduced Tg values in the ms were obtained by simulations only (ref 5), as well as the indication of NT-matrix covalent bonding. The text does not mention that this is a simulation result, and it should. There are various papers (e.g., Pötschke) that contradict these observations.

The articles that present simulation results are now pointed out specifically in the amended manuscript. Also, a number of references which present empirical results are now cited and added to the reference list.

Since the matrix-filler bonding is at the center of this opinion ms (L101-5), a much broader scope on relevant systems should be given (not only the amido-Amine functionalization-epoxy) to show that the concept is general.

Bearing in mind that this is a short opinion article (and not a review), I argue that even a small number of examples is sufficient to make a relevant point to express a hypothesis. Yet additional information on interfacial interactions can now be fined in the cited new references.

Regarding the colligative approach of the mechanical properties in composites mentioned in this ms: it has been shown in so many papers that most properties are almost always way below the predicted by the rule of mixture. I cannot see the relevance of such a statement to this ms.

I agree. the colligative approach is now mentioned only once where cited from an earlier paper.

In summary, the author keeps referring to his/her older papers, and one wonders where is the new opinion here? It is vaguely mentioned only in the last few lines of the ms. What are the other physical properties the author refers to? This is not clear at all.

The references to two of our previous papers are essential for the discussion and the build up of the hypothesis, which is now mentioned clearly in the Discussion and Conclusions sections.

Reviewer 2 Report

The manuscript aims to explain the point about the reason of nanoparticles (CNTs) reduce the glass transition temperature of nanocomposites.  Some important results have been obtained in this paper. However, the current version still needs to be improved as follows.

1. The introduction fails to answer the questions to explain the point about the reason of nanoparticles (CNTs) reduce the glass transition temperature of nanocomposites. The introduction needs to clearly identify the area of lack of knowledge for the specific topic. This will help better identification of the originality of the work about the opinion.

2. More research results of the last three years are needed to explain this opinion.

3. The manuscript should provide an in-depth discussion of nanoparticles (CNTs) reduce the glass transition temperature of nanocomposites between theory and practical.

Author Response

I wish to thank the reviewer for his/her useful comments which helped me improve the article. Here is my response point by point:

  1. A new paragraph has been added to the beginning of the Introduction, which explicitly explain why the article focuses on the specific case of lowering the glass transition temperature by nanoparticles.
  2. New references have been added and included in the discussion to explain the hypothesis stated by the paper.
  3. Done. The discussion has been broadened accordingly.

Round 2

Reviewer 1 Report

I have no further comments beside a couple of typos:

size. of the nanoparticles--> size of the nanoparticles

different then microcomposites --> different than microcomposites

Reviewer 2 Report

After the revisions and the explanations made at the request of the the referee, the manuscript is now of good quality and can bring interesting information to the community of the composites.